# Reproducibility Report
# RetinaFace: Single-shot Multi-level Face Localization in the Wild

## Reproducibility Summary

**Scope of Reproducibility**

RetinaFace [2] is a deep learning model that detects faces in images by proposing rectangular areas (bounding boxes) for every single face. Unlike the other current state-of-the-art models, this study proposes a multi-task loss calculation by also computing the coordinates of 5 facial landmarks (eyes, nose, and two sides of the mouth) and 3D face mesh with 1000 points concurrently. Additionally, the proposed model also adapts a cascaded structure [13] and deformable convolution layers (DCL) [1]. The scope of this paper includes the whole model structure excluding DCL. Additionally, The tasks implemented are limited only to face bounding box detection and landmark localization tasks, since the 3D point detection database is not publicly shared.

**Methodology**

For this challenge, I implemented this model in Julia programming language, by using the Knet deep learning framework. The whole model is implemented from scratch. There are official and unofficial implementations are available but these codes only contain a subset of the whole model proposed in the paper. In the context module part and for constructing the methods related to box proposal, these repositories are taken as examples. For training, the WIDER FACE database [12] is preferred and as landmark data, custom annotations created by the original paper's authors are used. Model is trained in one Tesla V100 GPU with a batch size of 10 for 60 epochs, which lasted approximately 9 days.

**Results**

The average precision (AP) metric is used for evaluation and the results are 0.093 lower in the Easy, 0.076 in the Medium, and 0.129 in the Hard subsets of WIDER FACE. Possible reasons for this performance difference are discussed in the *Limitations & Problems* section.

**What was easy**

Since the model only uses a small set of operations (convolution, batch normalization, unpooling, softmax, and ReLU). Therefore implementing the whole model was easy except for the loss calculation part.

**What was difficult**

The selection process of which box proposals are for faces and which are for background and how to balance their losses were not explained in the original paper in detail. Because of these obscurities, implementing the loss calculation was difficult.

**Communication with original authors**

I contacted them to request access to the 3D face points database but learned that that data belongs to a start-up company and is not publicly licensed.

# 1    Introduction

Face Detection is a crucial problem in Computer Vision. Depending on the position of the camera relative to the person itself, detecting faces may bring some obstacles such as occlusions, really small-sized face captures compared to the frame length, change of illumination, etc. To overcome these obstacles, some common structures are preferred in the current state-of-the-art detection models. First of all, a mechanism called Feature Pyramid Network (FPN) [4] is integrated for processing different-sized intermediate outputs retrieved from backbones such as ResNet [3] or VGG [10]. With this approach, each intermediate output learns to focus on faces with different scales, and hence the overall performance increases.

Secondly, two main ways are proposed for detection tasks: single-shot [5] and two-shot [8]. While the single-shot method focuses on predicting the coordinates only by feeding the input to the regarding model one time, the two-shot method predicts some intermediate outputs by feeding the input for the first time and more accurate bounding box proposals are found by using these intermediate outputs and feeding the input to the model a second time. Although two-shot models may compute more accurate results compared to single-shot models, longer processing time and extra computational load directed more recent studies to design single-shot models. With the recent development and new methods on single-shot object detection, current state-of-the-art models also achieved to outperform two-shot based approaches [11]. Currently, three single-shot detection models RetinaFace, ProgressFace [14] and HAMBox [6] are showing state-of-the-art performances in face detection. While HAMBox deals with adjusting the anchor boxes to have a greater intersection of union (IOU) values with the ground truth values, ProgressFace proposes a progressive learning model, which learns faces from large to small scales gradually.

# 2    Scope of Reproducibility

Current and previous state-of-the-art face detection models are bounded with using only background/face classification and bounding box regression. Other tasks such as facial landmark localization or 3D face points detection are ignored, which may improve the overall face detection performance if integrated into the model. Different from these studies, RetinaFace aims to benefit from landmark localization and 3D face point detection tasks to improve its performance further. Since 3D points data is not shared in public, this reproduction study is only limited with landmark localization task added to the main detection task.

In this study, I am testing:

- How much the model's performance increase if landmark task is also added?
- How much improvement is seen if a context module from SSH [7] is also integrated into the baseline model?
- Does the addition of the cascaded structure improves the model performance?

# 3    Methodology

The codes of this project are shared as a supplementary material and the GitHub repository can also be shared upon request.

## 3.1    Databases

This study uses only one face detection database: WIDER FACE Face Detection Database [12]. This database does not include any landmark annotation by default. Therefore, *84.6k* Faces in the WIDER FACE dataset are also manually annotated with 5 landmarks each[1] during the original research.

## 3.2    Architectural Details

For the implementation, Julia v1.5.3 is selected as the programming language and Knet v1.4.5 is preferred as the deep learning framework. Knet is a low-level framework compared to other popular deep learning frameworks such as PyTorch, TensorFlow, or Keras. It only includes operation functions required during the model implementation but does not provide objects for these operations. Therefore, even convolution and batch normalization (BN) layer objects are constructed manually. Furthermore, Knet does not have support for multi-GPU design. Hence, the implementation is only based on a single GPU usage.

---

[1]https://www.dropbox.com/s/7j70r3eeepe4r2g/retinaface_gt_v1.1.zip?dl=0

### 3.3 Inputs & Outputs

640x640 RGB images are used as the model inputs. For each of the images, confidence scores of each area proposals (one score for being background and one score for being a face), box coordinates (center coordinates, width, and height), and 5 facial landmark locations are retrieved as output.

### 3.4 Data Augmentation

As proposed in the original paper, the techniques below are used to augment the data:

- **Random Crops:** the images are cropped in square shapes randomly, then reshaped to 640x640 and the annotations are arranged concerning these crops.
- **Horizontal Flip:** With 0.5 probability, the images are flipped horizontally.
- **Color distortions:** With 0.5 probability for each, brightness, saturation, contrast, and hue distortions are applied to the input image.

Although data augmentation is also implemented, the different model variations are trained without any augmentation and the final fully structured model is also trained with augmentation to observe the impact of the augmentation process.

### 3.5 Model Description

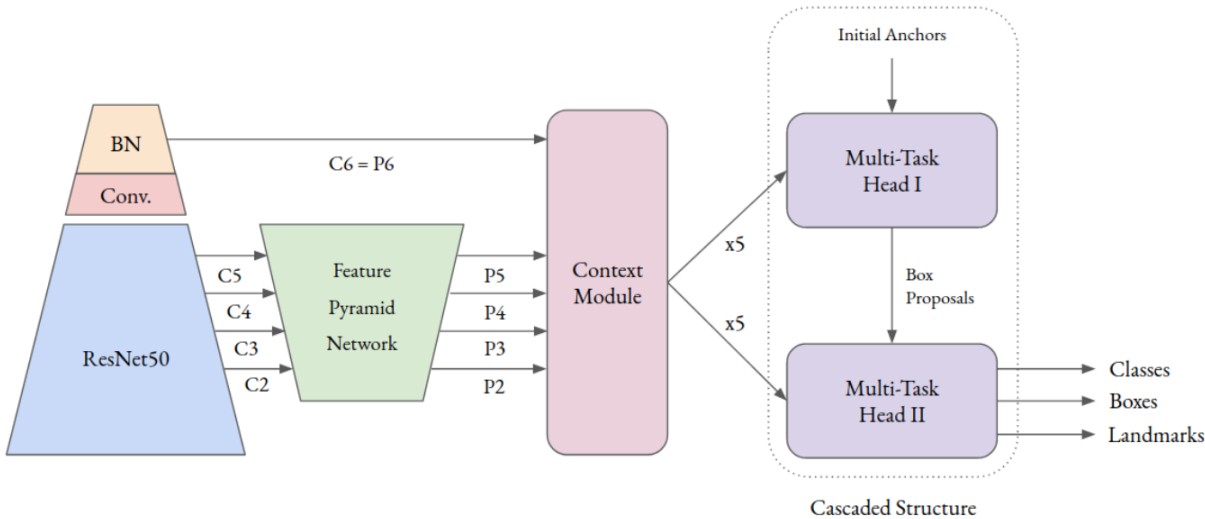

Figure 1: Structure of the Whole Model

**Backbone.** ResNet50 [3] is used with ImageNet pre-trained weights. From this structure, intermediate outputs of each block of convolutions are extracted. In total, there are 4 different outputs: C2, C3, C4 and C5 respectively. The network is fed with a batch of 640 x 640 RGB images in training and the dimensions below are given for each of these intermediate feature maps:

- C2: 160 x 160 x 256 x N
- C3: 80 x 80 x 512 x N
- C4: 40 x 40 x 1024 x N
- C5: 20 x 20 x 2048 x N

where N is the batch size. Furthermore, a 3x3 convolution + batch normalization layer (from now on, all convolution + batch normalization layer blocks will be called as ConvBn) with the stride size 2 and the filter size of 256 is also defined additionally on top of C5. The parameters of this layer are initialized with the Xavier method[2]. With this additional

---

[2]https://denizyuret.github.io/Knet.jl/latest/reference/#Knet.Train20.xavier

layer, an extra output called C6 is created with a size of 10 x 10 x 256 x N.

**Feature Pyramid Network.** After retrieving the outputs C2-C5, all of these values are passed on 1x1 ConvBn layers to reduce their third dimensions to 256. The new outcomes are named P2-P5. Starting from the topmost feature map (P5), an unpool (upsampling) operation is applied to equalize the first and second dimensions of the upper and lower feature maps. Then the unpooled upper layer and the lower layer are added together. Lastly, the outcome is passed to an additional ConvBn structure with a kernel size of 3x3. C6 is excluded from all of these processes. The latest values of the intermediate feature maps can be summarized as:

- P6 = C6 (has the size: 10 x 10 x 256 x N)

- P5 = ConvBn_1x1(C5) (has the size: 20 x 20 x 256 x N)

- P4 = ConvBn_3x3(ConvBn_1x1(C4) + unpool(P5)) (has the size: 40 x 40 x 256 x N)

- P3 = ConvBn_3x3(ConvBn_1x1(C3) + unpool(P4)) (has the size: 80 x 80 x 256 x N)

- P2 = ConvBn_3x3(ConvBn_1x1(C2) + unpool(P3)) (has the size: 160 x 160 x 256 x N)

where each of the ConvBn layers is defined independently from each other. From now on, all of the P2-P6 outputs will be referred to as lateral paths.

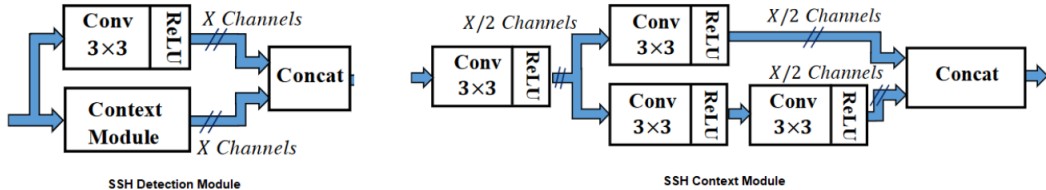

Figure 2: SSH Detection Module [7]

**Context Modules.** Each of the lateral paths is passed into independent context modules, which have the same model structure but different weights (in total, there are 5 context modules, one for each lateral path). The context module design is directly adopted from SSH Detection Module [7] and it can be further investigated in Figure 2. Additionally, batch normalization layers are added after each of the convolution layers in the figure. The input size is preserved by adding padding of size 1 to each convolution. All of the lateral paths are updated with this module.

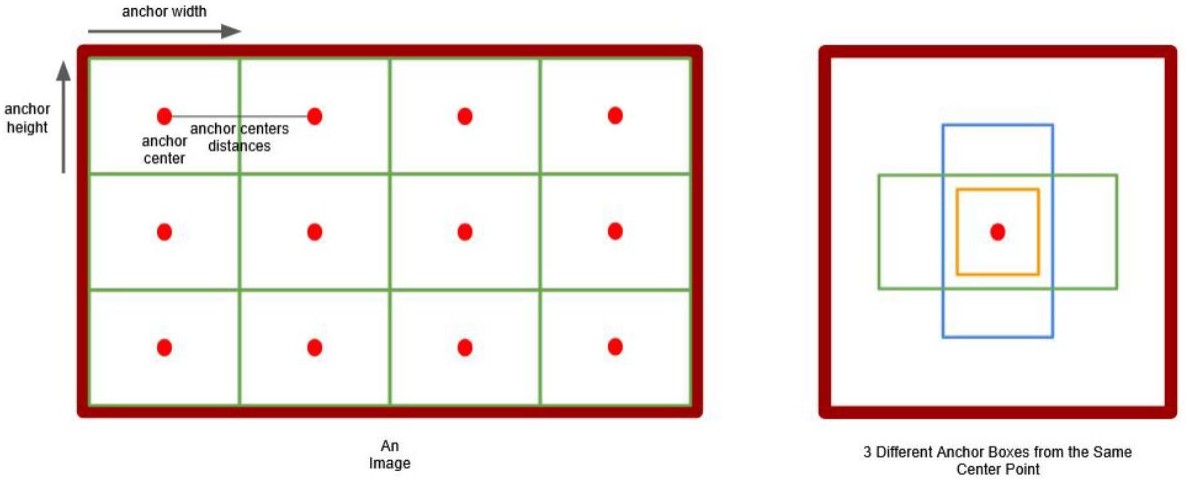

Figure 3: Anchor Box Demonstration

**Anchor Boxes.** Anchor boxes are predefined rectangles that divide the image into grids with different sizes. They have basically a height, width, and two center coordinates (x and y). Figure 3 demonstrates how anchor boxes are located. There is a constant distance between the centers of each neighboring anchor box pairs. Also, more than one anchor boxes can be constructed from the same center point. Each anchor box includes 4 values (center x, center y, width, and height).

The size of the model output is equal to the size of the combination of anchor boxes. In this specific implementation, there are 102,300 anchor boxes for a single image, the final outputs of the model are 2 x 102,300 for classification, 4 x 102,300 for box proposal, and 10 x 102,300 for landmark localization tasks. By calculating the IOU values of each of the anchor boxes with each of the ground-truth bounding boxes and by selecting the maximum-IOU-value-giving anchor indices per ground truth face, the indices in the model output which are responsible for predicting the bounding box values of the ground truth objects can be determined.

Since faces usually share similar height and width ratios, selecting anchor boxes that have 1:1 height and width ratios would be logical. In RetinaFace, each lateral path output includes W x H many anchor box centers if the dimension of the lateral output is W x H x C x N. Furthermore, each anchor box center is responsible for 3 different anchor boxes, all having 1:1 width and height ratios. The details of each lateral outputs are as follows:

| Laterals | Anchor Center Sizes | Center Distances (in Pixels) | Edge Lengths (in Pixels) | | |
|:---:|:---:|:---:|:---:|:---:|:---:|
| P6 | 100 | 64 | 256 | 322.54 | 406.37 |
| P5 | 400 | 32 | 128 | 161.26 | 203.19 |
| P4 | 1600 | 16 | 64 | 80.63 | 101.59 |
| P3 | 6400 | 8 | 32 | 40.32 | 50.80 |
| P2 | 25600 | 4 | 16 | 20.16 | 25.40 |
| Total Centers | 34100 | Total Anchors | 102300 | | |

Table 1: Anchor Data for Each Lateral Path

**Multi-Task Heads.** After passing the context modules, the lateral paths are put 1x1 convolutions to convert their third dimension size to the correct output format. Assuming that W x H x C x N is the size of a lateral path after its corresponding context module and each anchor center is responsible for "A" many anchors (A is selected as 3 in the original paper), the final outputs retrieved from only one lateral path should have the dimension:

- Classification Output: $W$ x $H$ x $2A$ x $N$
- Bounding Box Output: $W$ x $H$ x $4A$ x $N$
- Landmark Output: $W$ x $H$ x $10A$ x $N$

Multi-task heads convert the dimension C to required sizes by 1x1 convolutions. Each lateral path has its own classification, bounding box, and landmark head and after this process, each lateral path creates 3 different outputs. However, the processes after multi-task heads are common to each lateral path. Therefore, these outcomes are concatenated. To achieve this, classification outputs are reshaped to the size 2 x (W · H · A) x N, bounding box outputs to 4 x (W · H · A) x N and landmark outputs to 10 x (W · H · A) x N. Afterward, the results of each lateral path are concatenated along their second dimensions. In the end, only 3 outputs are constructed:

- Classification Final Output: 2 x 102300 x N
- Bounding Box Final Output: 4 x 102300 x N
- Landmark Final Output: 10 x 102300 x N

where (W · H · A) is the number of anchor boxes responsible for one lateral path output, 102,300 is the total number of anchor boxes.

**Prediction & Ground Truth Conversions.** The predicted box and landmark coordinates are designed to be scale invariant. Therefore, some transformation equations are applied to convert ground truth data to predicted value format. For any ground truth with a center coordinate $(x_c, y_c)$, with the lengths $(w, h)$ and with a landmark point $(x_l, y_l)$:

- $x_c^p = (x_c - x_c^a)/w^a$
- $y_c^p = (y_c - y_c^a)/h^a$
- $w^p = ln(w/w^a)$
- $h^p = ln(h/h^a)$
- $x_l^p = (x_l - x_c^a)/w^a$
- $y_l^p = (y_l - y_c^a)/h^a$

where $(x_c^p, y_c^p)$ corresponds to the box center in the prediction format, $(w^p, h^p)$ to the box lengths in the prediction format, and $(x_l^p, y_l^p)$ to the landmark point in the prediction format. Additionally, $(x_c^a, y_c^a)$ is the center coordinate and $(w^a, h^a)$ are the lengths of the corresponding anchor box.

**Online Hard Example Mining (OHEM) [9] & Loss Calculation.** The training process utilizes both classification and regression losses. For classification, the negative log-likelihood (NLL) is preferred and for regression, the smooth L1 loss is selected as proposed in the original paper.

$$\text{smooth}_{L_1}(x) = \begin{cases} 0.5x^2 & \text{if } |x| < 1 \\ |x| - 0.5 & \text{otherwise,} \end{cases}$$

The regression losses are computed only from the positive anchors (the ones that are matched with a ground-truth object) since there are no values to regress for backgrounds. The box losses are computed by using width, height, and center coordinate values. The overall loss structure for a single selected index $i$ is given below:

- $L_i = L_{cls}(p_i, p_i^*) + p_i^* L_{box}(t_i, t_i^*) + p_i^* L_{pts}(l_i, l_i^*)$
- $L_{cls}(p_i, p_i^*) = -ln(1 - |p_i - p_i^*|)$
- $L_{box}(t_i, t_i^*) = smooth_{L1}(|t_i - t_i^*|)$ for each $(x_c, y_c, w, h)$
- $L_{pts}(l_i, l_i^*) = smooth_{L1}(|l_i - l_i^*|)$ for each $(x, y)$ of 5 landmarks

where $p_i^*$ is 1 if the proposal belongs to a ground-truth face and 0 otherwise, $p_i$ means the probability of a bounding box proposal to be positive, $t_i^*$ is a ground truth box value converted as explained in the *Prediction & Ground Truth Conversions* subsection, $t_i$ is a box value prediction, $l_i^*$ is a ground truth landmark value converted in a scale-invariant style, $l_i$ is a landmark value prediction.

There is a significant difference in terms of counts of the positive anchors and negative anchors (the ones that are not assigned to a specific ground-truth object). Therefore, the OHEM method is used to balance the ratios of positive and negative anchors selected.

After calculating the output of the multi-head module, the IOU values are calculated between each ground truth object and each anchor box of the corresponding multi-head module. Then, the maximum IOU value is calculated for each of the anchor boxes. The anchor boxes that have an IOU value bigger than the positive threshold are selected as positive anchors. The positive threshold for the first multi-task module is set as 0.7 and for the second multi-task module as 0.5.

Similarly, the ones among the non-positive anchors that have a maximum IOU less than the negative threshold with any of the ground truth object are chosen as negative anchor candidates. The negative threshold is assigned as 0.3 in the first multi-task head and 0.4 in the second multi-task head. According to the OHEM rule, the number of selected negative anchors must be equal to at most 3 times the number of positive anchors. Hence among the candidates, a subset having the greatest NLL loss values is selected as negative anchors.

**Cascaded Structure.** Instead of loading the whole final proposal job to only one multi-task head structure, a cascaded model divides this job into 2 multi-task head modules. Same as before, the model retrieves lateral outputs until the context module. Then by using already-existing static anchor boxes and the first multi-task head module, it produces classification scores, bounding box, and landmark predictions. A loss from this process is also calculated. Then, instead of using the actual static anchor boxes, the model uses the bounding box predictions calculated from the first multi-task head module as anchors and applies the same process this time with the second multi-task head module. The outcomes

of the second multi-task head become the final outputs of the model and the sum of the losses calculated from both multi-task head modules are added for computing the final loss.

### 3.6 Computational Requirements

The code is tested on both Windows and Linux and confirmed that it is fully functioning. To obtain the best performance, Tesla V100 GPU is suggested since it can run up to 10 images per batch. The model is also tested in Tesla T4 and it is seen that batch size can be set at most 4. The full model completes 6.5 epochs per day if the batch size is set as 10 and discarding the cascaded structure increases this amount to 10 epochs per day.

## 4 Experiments & Results

### 4.1 Hyper-Parameters

As it is given in the actual paper, the SGD optimizer (momentum: 0.9, weight decay: 0.0005) is selected for training purposes. Since this implementation remained limited with single GPU usage, the batch size is selected as 10, which is the highest amount of image count in a batch that a Tesla V100 GPU memory supports while training.

The learning rate is set to 0.001 between the 1st and 27th epochs, 0.004 between 28th and 39th, 0.001 again between 40th and 49th, and 0.0001 between 50th and 60th. In total, each of the model variations is trained for 60 epochs and the individual checkpoints are chosen as final, where the highest scores are achieved. The learning rate of 0.01 is not used although it is preferred in the original paper, because the batch size is 3 times lower in this implementation compared to the original training batch size, and selecting high learning rates may cause unstable updates.

### 4.2 Different Model Variations

To test the hypotheses mentioned in the *Scope of Reproducibility* section, different variations of the model are trained separately:

- **Baseline:** ResNet50 + FPN + Landmark Localization Task

- **Context Module without Landmark:** ResNet50 + FPN + Context Module

- **Context Module with Landmark:** ResNet50 + FPN + Context Module + Landmark Localization Task

- **Full Model:** ResNet50 + FPN + Context Module + Landmark Localization Task + Cascaded Structure

### 4.3 Evaluation of AP in WIDER FACE Validation Data

In the actual paper, the evaluation also included the performance of the landmark localization task but in this reproduction study, the evaluation scope is limited only to bounding box prediction performance and the average precision (AP) metric used for this purpose. AP is calculated by taking the IOU threshold as 0.5 and iterating through 1000 steps of confidence levels between 0 and 1. The models are evaluated only with WIDER FACE validation data and this data is separated into 3 groups (Easy, Medium, and Hard) concerning their difficulty. The confidence threshold is set to 0.02 to decrease the total computation time, then the top 5000 predictions are selected among the candidates and lastly, the non-maximum suppression (NMS) method is applied with a threshold of 0.4 to eliminate redundant area proposals.

In table 2, the results of the original RetinaFace model, other state-of-the-art models, and my different model variations are provided. While the HAMBox model achieves the highest performance in all of the 3 subsets of WIDER FACE validation data, RetinaFace performs close to HAMBox.

The best-performing model results in 0.093 lower AP value in the Easy subset, 0.076 lower in the Medium, and 0.129 lower in the Hard subset compared to the original paper results. As mentioned in the *Scope of Reproducibility* section, the model performs better when landmark localization task is included or the Context Module is also added. However, adding a cascaded structure causes a performance drop in contrast to our expectations. Although the reason for the negative effect of the cascaded structure is not clear, the possible main reasons for the performance difference between the original paper and this implementation are discussed in the following *Limitations & Problems* section.

| Model | WIDER FACE Easy | WIDER FACE Medium | WIDER FACE Hard |
|---|---|---|---|
| HAMBox [6] Baseline | 0.943 | 0.931 | 0.894 |
| HAMBox Final | **0.970** | **0.964** | **0.933** |
| ProgressFace [14] | 0.968 | 0.962 | 0.918 |
| RetinaFace Baseline | 0.958 | 0.952 | 0.899 |
| + Context Module with DCL | 0.961 | 0.956 | 0.903 |
| + Cascade | 0.962 | 0.957 | 0.906 |
| + Landmark Loss | 0.966 | 0.959 | 0.912 |
| Baseline | 0.842 | 0.864 | 0.769 |
| Context Module without Landmark | 0.865 | 0.878 | 0.767 |
| Context Module with Landmark | 0.873 | 0.883 | 0.783 |
| Full Model | 0.842 | 0.854 | 0.752 |

Table 2: WIDER FACE Validation Data AP Scores

## 4.4 Some Example Visual Results

These results are retrieved from the best performing model variation. For the prediction, the NMS threshold is set to 0.2 and the confidence threshold is set to 0.5. Images are taken from WIDER FACE validation data. If the faces are not too small, then the model mostly detects the faces (Figures 4 and 5) even when there is a bad lighting (Figure 6) or the faces from a slightly different domain (a drawing in the case of Figure 7). On the other hand, if the faces are too small or the resolution is not very clear (Figures 8, 9, and 10), then the model may miss the faces in the picture.

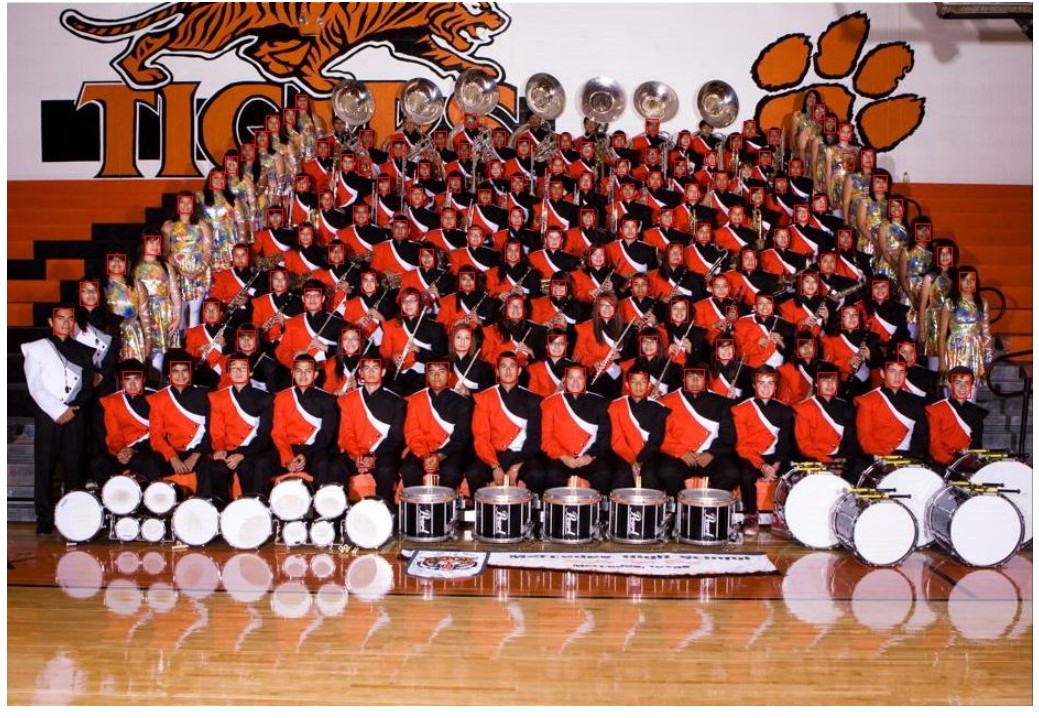

Figure 4: Good Example Result I

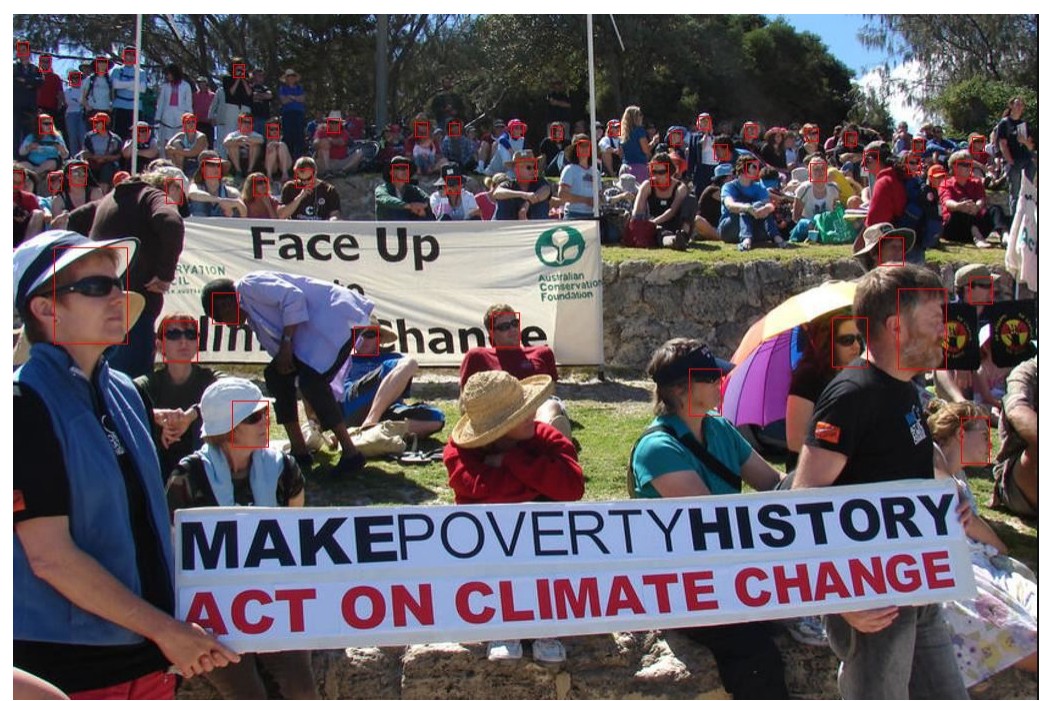

Figure 5: Good Example Result II

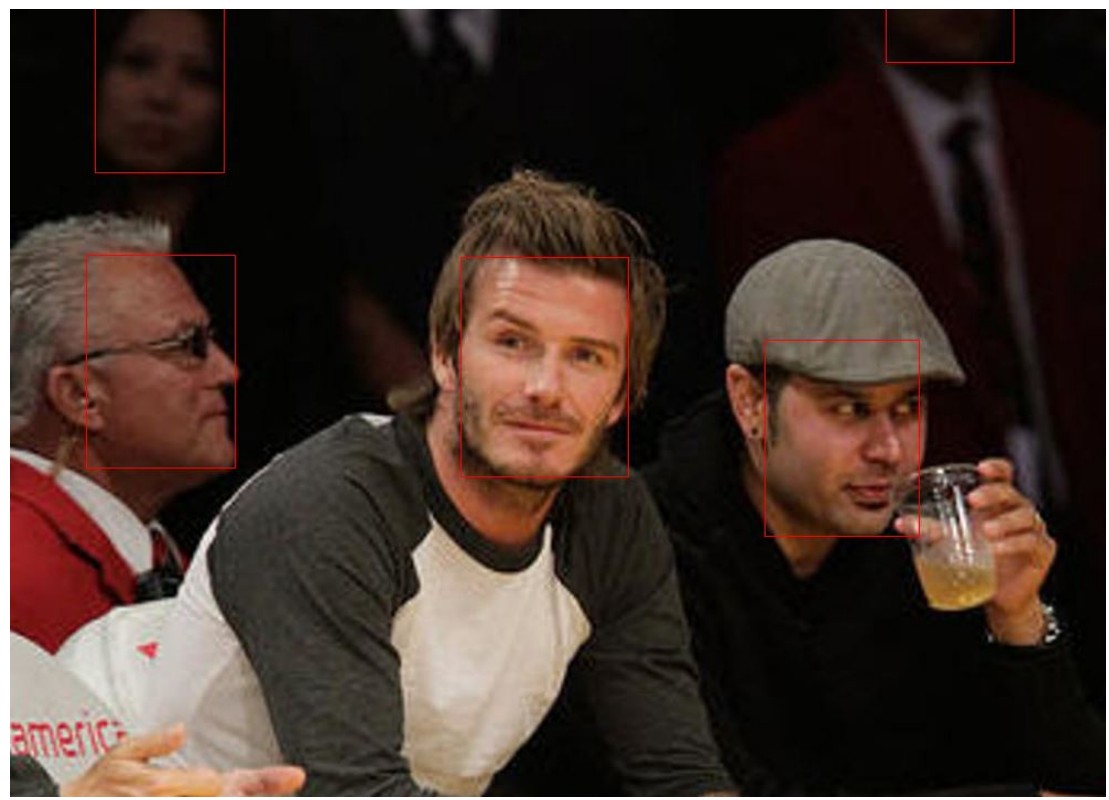

Figure 6: Good Example Result III: Bad Lighting

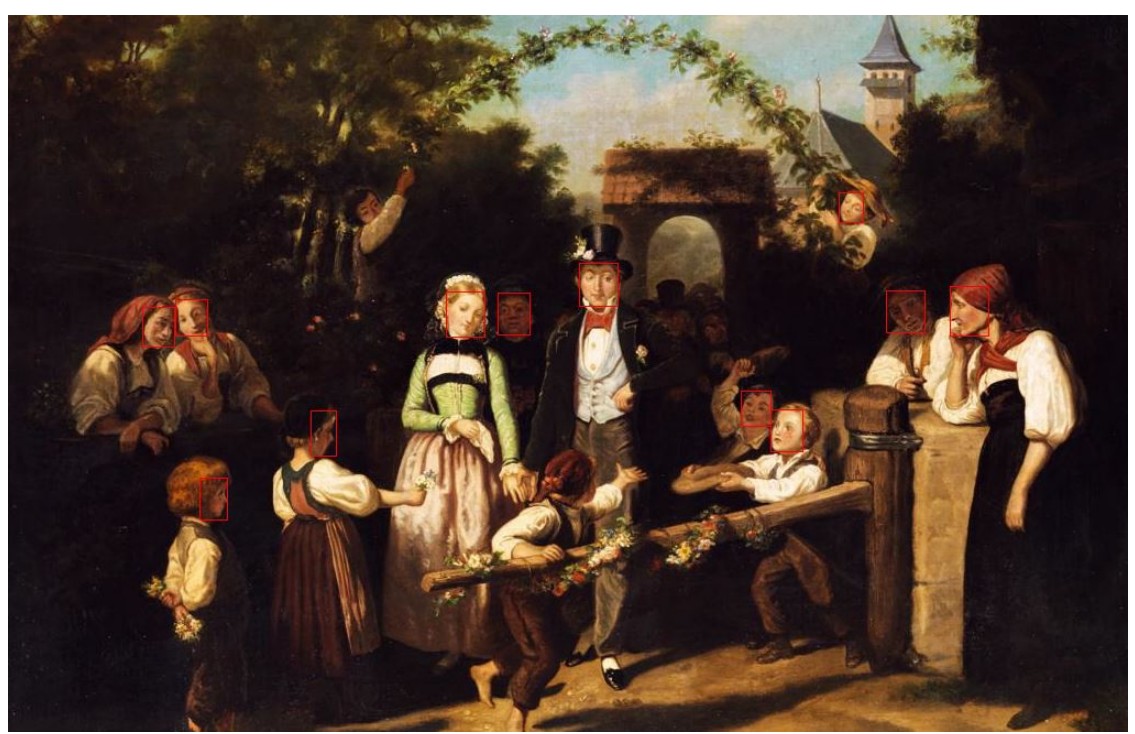

Figure 7: Good Example Result IV: Drawing Domain

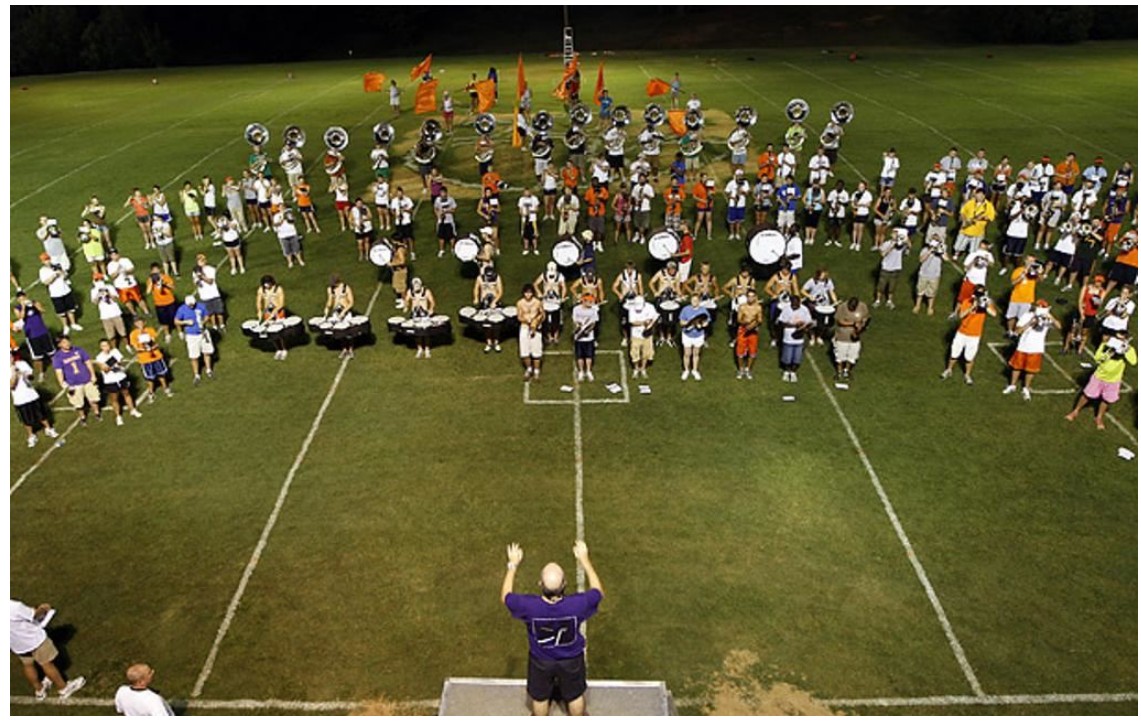

Figure 8: Bad Example Result I: Small Faces, No Faces Detected

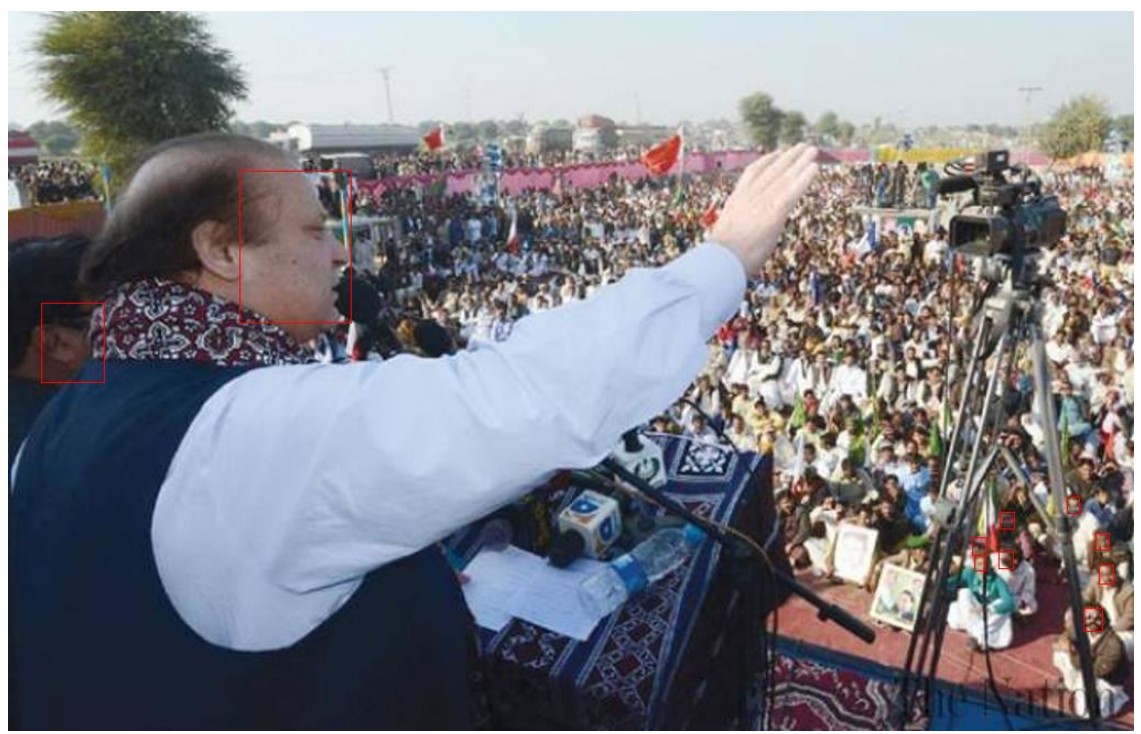

Figure 9: Bad Example Result II: Not a Good Resolution

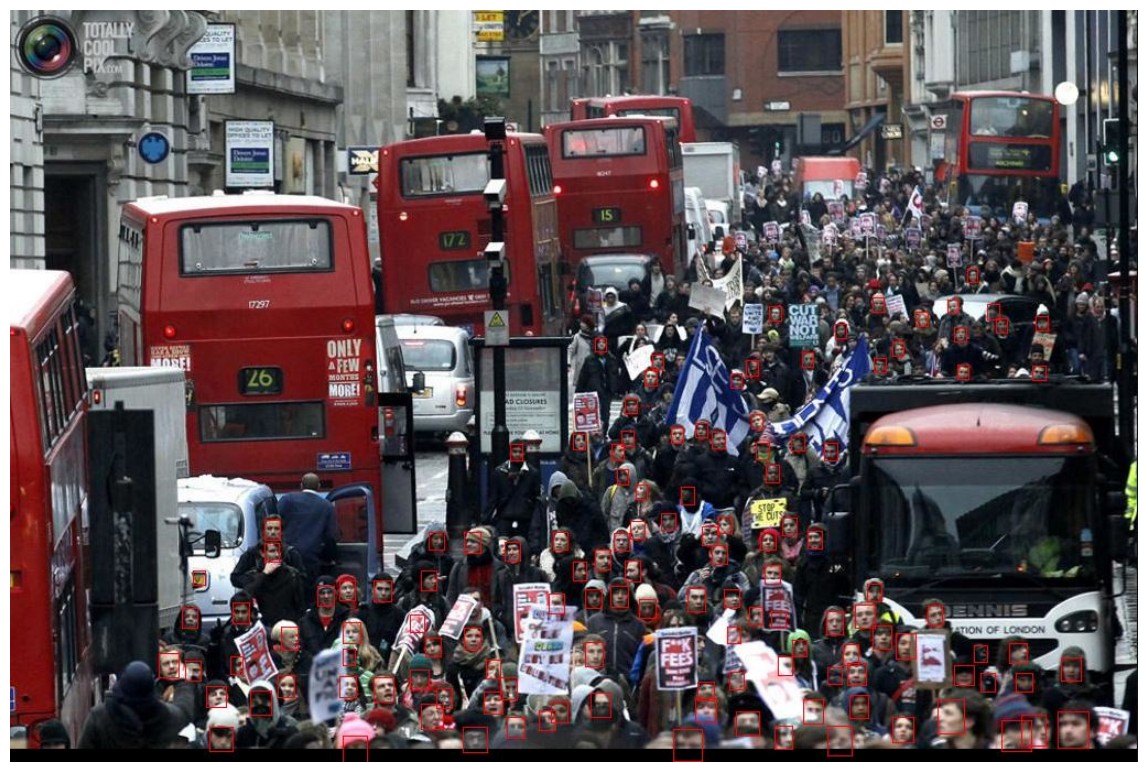

Figure 10: Bad Example Result III: Small Faces

# 5   Limitations & Problems

In this section, some limitations of the framework and the main reasons for the difference between the actual paper's results and this implementation's results will be discussed.

**Lack of Deformable Convolution Layers.** The actual paper uses Deformable Convolutions instead of regular convolutions for the Context Module structure. This special type of convolution achieves to "learn the offsets from the target tasks, without additional supervision" [1]. Therefore, it increases the overall performance of the model. However, this type of layer structure is not available in Julia or Knet, and implementing this structure requires a significant amount of additional work. Thus, this improvement is left as future work.

**Lower Batch Size.** The original RetinaFace model uses 4 V100 GPUs and in total 32 images as a batch. However, Knet does not fully support training models on multiple GPUs and the maximum number of images a V100 GPU can take are 10 images. Hence, the model is limited to complete its training with 3 times lower batch size. Having a lower batch size also causes more unstable updates on the model. Therefore, it becomes harder to find the optimal checkpoint and the chance of finishing the training with sub-optimal local minimum increases.

**Knet Defaults During Learning Rate Change.** During training, each of the trainable parameters stores an additional optimizer field, where the optimizer name and its specific parameters are stored. With the current Knet configurations once this field is set, calling another optimizer with different parameter settings does not change the optimizer field of the parameters unlike the other frameworks such as PyTorch or Tensorflow. Hence, the model continues to be trained with the initial optimizer and learning rate until the end. To change the learning rate or the optimizer, each of the optimizer fields of each trainable parameter has to be set to *"nothing"*. I realized this problem in the last couple of days of the first submission and therefore, I could only submit some results that are significantly lower than the original paper.

**GPU-CPU Data Transfer.** While recording the gradient flow of the training batch, Knet also forbids to slice the final outputs and compute a loss from these sliced sub-parts when the data is on GPU. Therefore, a constant data transfer between the GPU and CPU takes place in this implementation during the loss calculation. This deficiency also increases the overall run-time.

# 6   Conclusion

In this paper, the structure of the RetinaFace model is analyzed, the implementation process is explained, the configurations and different experiments are shared and the results are discussed. The scope of the reproducibility is defined as testing if the model's performance increases when landmark localization task, context module, and/or cascaded structure are included in the model's structure.

According to the AP results retrieved for different model variations, it is shown that adding context module and landmark task to the model increases the performance. However, including the cascaded structure resulted in a decrease in the overall performance. The evaluation score difference between the original paper and this implementation is between 0.076 and 0.129 but the lower batch size and lack of deformable convolutions are possible reasons for this performance drop. Overall, the model mostly detects faces when they are not extremely small in the given image.

# 7   Discussion

## 7.1   What Was Easy

As stated in the summary page and the model description, RetinaFace only uses a couple of layer and activation structures. Excluding the deformable convolution layers used in the original model, it is easy to implement the whole model structure (except the loss calculation) even with a low-level deep learning framework like Knet.

## 7.2   What Was Difficult

Implementing the loss calculation structure was the hardest task because of the lack of information in the original paper. No detailed explanation was given to select the positive and negative anchors, OHEM technique was stated to balance the negative and positive anchor selection but no extra hyper-parameters and instructions to use the OHEM method were provided. To overcome this issue, official and unofficial implementation codes and the issues opened in these repositories are checked. Additionally, the OHEM paper is read and blogs on that issue are visited.

Implementing the AP metric in Julia was also a challenging task because it is a complicated metric and for this specific case, 3 different AP evaluations have to be made for each of the subsets of WIDER FACE validation data: Easy, Medium, and Hard. These subsets are not separated in terms of image files but are created by selecting a subset of faces for every single image. To avoid any false evaluation results, I used a python repository[3] which includes all of the AP evaluation processes for every single subset of the validation data.

## 7.3 Communication with Original Authors

I only communicated with the authors to request the 3D facial points data to extend my model with the 3D point prediction task. However, they indicated that the data is not licensed public. Therefore, I excluded the 3D point detection task from the model structure. Other than this, the model was explained mostly successfully in the original paper and it was enough for me to implement most of the parts.

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
