# OpenReview forum: "Reproducibility Report RetinaFace: Single-shot Multi-level Face Localization in the Wild"
_ML_Reproducibility_Challenge/2020 — Reject_

### Official Review · AnonReviewer3 · 2021-02-21
**Good effort in reproducing the results but incoherent discussion.**

**Rating:** 5
**Confidence:** 3

**Review:**

The paper is poorly written with an open-ended conclusion and the results are not exhaustive. Also, there is a lot of ambiguity around reproducibility since the author found a bug in their code. They state to provide the full results after resolution.

**Familiar With The Original Paper:**

I have read the original paper

**Reproducibility Summary:**

Report has summary

---

### Official Review · AnonReviewer2 · 2021-03-01
**with implementation details**

**Rating:** 4
**Confidence:** 3

**Review:**

The work of RetinaFace is reproduced here, to address the long standing problem of face detection, and evaluated on the dataset of WIDER. The reproduced method achieves on average 11%-18% worse than the original reported results, which seems to due to a bug in the reproducing effort?? This important aspect is not clear.   The bright side is this work comes with detailed implementation description.


**Familiar With The Original Paper:**

I have not read the original paper

**Reproducibility Summary:**

Report has summary

---

### Official Review · AnonReviewer1 · 2021-03-09
**Solid implementation in Julia.**

**Rating:** 7
**Confidence:** 3

**Review:**

The submission implemented RetinaFace using Knet framework. It contains most of the features from the original paper. In particular, the author conducted some ablation studies focusing on landmark detection, context module, and cascaded structure. The results are discussed in detail in the report.

The strength of the submission is its solid, from-scratch Julia implementation. Julia has gained lots of popularity in the data science community. But there are relatively fewer off-the-shelf models for state-of-the-art computer vision models. The submission is a good example that can be referenced by people who want to study implementation object detectors in Julia.

The performance is lower than the numbers reported in the original paper. The author attributed this to a bug in the Knet framework. The author observed some progress in training after the bug is fixed but could not finish the training in time. Due to this reason, I feel reluctant to give higher scores, but at the same time encourage to have this submission accepted due to its language & framework choices and in-depth discussion about the experiments.

**Familiar With The Original Paper:**

I have read the original paper

**Reproducibility Summary:**

Report has summary

---

### Decision · Program_Chairs · 2021-03-31

**Decision:**

Reject

**Comment:**

Overall reviews and/or the paper content not good enough for the AC to recommend to the journal.